# Strontium Substituted β-Tricalcium Phosphate Ceramics: Physiochemical Properties and Cytocompatibility

**DOI:** 10.3390/molecules27186085

**Published:** 2022-09-18

**Authors:** Inna V. Fadeeva, Dina V. Deyneko, Anna A. Forysenkova, Vladimir A. Morozov, Suraya A. Akhmedova, Valentina A. Kirsanova, Irina K. Sviridova, Natalia S. Sergeeva, Sergey A. Rodionov, Irina L. Udyanskaya, Iulian V. Antoniac, Julietta V. Rau

**Affiliations:** 1A.A. Baikov Institute of Metallurgy and Material Science RAS, Leninskie, 49, 119334 Moscow, Russia; 2Department of Chemistry, Lomonosov Moscow State University, 1, Leninskie Gory, 119991 Moscow, Russia; 3Laboratory of Arctic Mineralogy and Material Sciences, Kola Science Centre, Russian Academy of Sciences, 14 Fersman Str., 184209 Apatity, Russia; 4Herzen Moscow Research Institute of Oncology—Branch of the Federal State Budgetary Institutio, National Medical Research Center for Radiology of the Ministry of Health of Russia, 2nd Botkinsky Pr-d, 3, 125284 Moscow, Russia; 5Academician Yarygin Department of Biology, Federal State Autonomous Educational Institution of Higher Education Russian National Research Medical University Named after N.I. Pirogov, Str. Ostrovityanova, 1, 117997 Moscow, Russia; 6N.N. Priorov National Medical Research Center of Traumatology and Orthopaedics, 10 Priorova Str., 127299 Moscow, Russia; 7Department of Analytical, Physical and Colloid Chemistry, Institute of Pharmacy, I.M. Sechenov First Moscow State Medical University, Trubetskaya 8, Build. 2, 119991 Moscow, Russia; 8Department of Metallic Materials Science and Physical Metallurg, University Politehnica of Bucharest, Street Splaiul Independentei No 313, Sector 6, 060042 Bucharest, Romania; 9Istituto di Struttura della Materia, Consiglio Nazionale delle Ricerche (ISM-CNR), Via del Fosso del Cavaliere, 100-00133 Rome, Italy

**Keywords:** ceramic, strontium, tricalcium phosphate, strontium substituted tricalcium phosphate, strontium doped tricalcium phosphate, dissolution, cells adhesion, cytocompatibility

## Abstract

Sr^2+^-substituted β-tricalcium phosphate (β-TCP) powders were synthesized using the mechano-chemical activation method with subsequent pressing and sintering to obtain ceramics. The concentration of Sr^2+^ in the samples was 0 (non-substituted TCP, as a reference), 3.33 (0.1SrTCP), and 16.67 (0.5SrTCP) mol.% with the expected Ca_3_(PO_4_)_2_, Ca_2.9_Sr_0.1_(PO_4_)_2_, and Ca_2.5_Sr_0.5_(PO_4_)_2_ formulas, respectively. The chemical compositions were confirmed by the energy-dispersive X-ray spectrometry (EDX) and the inductively coupled plasma optical emission spectroscopy (ICP-OES) methods. The study of the phase composition of the synthesized powders and ceramics by the powder X-ray diffraction (PXRD) method revealed that β-TCP is the main phase in all compounds except 0.1SrTCP, in which the apatite (Ap)-type phase was predominant. TCP and 0.5SrTCP ceramics were soaked in the standard saline solution for 21 days, and the phase analysis revealed the partial dissolution of the initial β-TCP phase with the formation of the Ap-type phase and changes in the microstructure of the ceramics. The Sr^2+^ ion release from the ceramic was measured by the ICP-OES. The human osteosarcoma MG-63 cell line was used for viability, adhesion, spreading, and cytocompatibility studies. The results show that the introduction of Sr^2+^ ions into the β-TCP improved cell adhesion, proliferation, and cytocompatibility of the prepared samples. The obtained results provide a base for the application of the Sr^2+^-substituted ceramics in model experiments in vivo.

## 1. Introduction

Currently, extensive research and developments of new materials for bone restoration are carried out [1,2]. The most common of these materials are based on calcium phosphates (CP) and are similar to the mineral components of bone tissue, being also non-toxic, biocompatible, and osteoinductive [3,4]. CP-based materials can be used for cements [5], ceramics [6], and composites preparation [6,7].

CP-based ceramic is the most common material for filling bone defects [6], especially in places bearing a mechanical load [8]. Such orthophosphates as hydroxyapatite (HA, Ca_10_(PO_4_)_6_(OH)_2_) [9], tricalcium phosphate (β-TCP, β-Ca_3_(PO_4_)_2_) [4], brushite (DCPD, CaHPO_4_·2H_2_O) [10], and octacalcium phosphate (OCP, Ca_8_(HPO_4_)_2_(PO_4_)_4_·5H_2_O) [11] can be used for ceramics preparation. However, β-TCP-based ceramics are more applicable for bone tissue restoration, since β-TCP is more soluble than HA [12] and has the ability to be resorbed in the body [13,14]. Moreover, calcium phosphates are excellent accommodators of substitutions by foreign ions, favoring the restoration of the mineral component of the bone, and β-TCP ceramics appear to be very promising [13,15]. The efficiency of β-TCP for bone tissue restoration is comparable to that of auto- and allografts [16,17]. The review [16] presents a comparison of clinical trials of β-TCP, two-phase β-TCP/HA materials, HA cement, and autografts. Analysis of the clinical trial results demonstrated that β-TCP materials promote bone tissue repair better than HA and are similar to those of autografts [16]. A detailed comparison of the effectiveness of β-TCP and autografts was shown in [17], where the formation of new bone tissue during tooth transplantation was studied in vivo in mice [17]. A porous β-TCP scaffold and a porous β-TCP scaffold seeded with mouse bone marrow cells were implanted in animals. Parameters of bone volume, bone surface, and indicators of the trabeculae growth demonstrated no significant differences between β-TCP and β-TCP with bone marrow cells [17].

The possibility of the substitution of calcium ions into the β-TCP structure opens a way of modifying its properties [15,18]. Substitution ions can act as bioactive agents imparting antimicrobial properties or enhancing tissue regeneration processes [15,19,20,21]. From this point of view, a number of substituted CP materials containing Cu^2+^ [22], Mn^2+^ [23], Zn^2+^ [24], Fe^3+^ [25], and Gd^3+^ [26] ions have been studied.

Additionally, a number of investigations have been devoted to strontium (Sr) functionalization into calcium phosphates [27,28], since it is known to be effective for osteoporosis treatment [29]. Special interest in Sr is due to its properties that influence osteogenesis in vivo [20,30,31,32]. Furthermore, Sr^2+^ ions stimulate the growth of osteoblasts promoting new bone growth, while inhibiting the activity of osteoclasts and reducing the rate of bone resorption [27,33,34]. Furthermore, Sr stimulates angiogenesis, which additionally affects bone repair [21,27]. Many studies on cellular and animal models have been devoted to the bioactive effect of Sr [28,35].

It has been shown that isomorphic substitution Ca^2+^ → Sr^2+^ covers a wide range of *x* in Ca_3−*x*_Sr*_x_*(PO_4_)_2_ solid solutions with the β-TCP-type structure [36]. The limit of phase formation with the β-TCP-type structure corresponds to *x* = 2.286 [36]. This value is quite high in comparison with other divalent elements (M^2+^), such as Zn^2+^, Mn^2+^, and Mg^2+^, where the isomorphic substitution Ca^2+^ → M^2+^ can be realized up to *x* = 0.286 in Ca_3−*x*_M*_x_*(PO_4_)_2_ [37]. The excess of Sr^2+^ ions in the β-TCP host leads to the formation of a palmierite-type phase [38], which is not suitable as a bioactive material. The difference in the substitution level of Sr^2+^ and M^2+^ ions is due to the ionic radius (the incorporation into different crystal sites of the initial β-TCP host) and chemical similarity in comparison with the Ca^2+^ ions [37].

The physiochemical aspects of the Sr^2+^ substitution in CPs were previously considered in [39,40,41]. Some recent studies presented data on Sr^2+^-substituted HA [42] and α-TCP [43] phases or biphasic materials [44]. However, materials based on the above-mentioned CPs have several significant disadvantages. HA is slowly resorbed in the body environment and thereby delays the formation of native tissue [45]. The α-TCP phase is thermodynamically unstable, easily passes into calcium deficient HA, and, therefore, does not guarantee a single-phase product during use [46].

At the same time, there is a number of studies dedicated to the Sr^2+^-substituted β-TCP (SrTCP). The effect of Sr^2+^ substitution in the β-TCP structure, phase composition, and powder microstructures was studied in [47,48,49,50,51,52]. The mechanism of Sr^2+^ substitution into β-TCP was previously shown in [53]. SrTCP ceramics and the influence of Sr^2+^ concentration on the sintering of ceramics were investigated in [54,55].

Due to the outstanding properties of Sr^2+^-containing CPs and their ability to influence osteogenesis in vivo, there is a need to expand the data regarding the Sr^2+^ substituted β-TCP ceramics. In the present work, the Sr^2+^-substituted β-TCP ceramics were obtained with varying Sr^2+^ concentrations, and their behavior in a standard saline solution was investigated. The dissolution process of the ceramic samples accompanied by the formation of different phases was analyzed by means of powder X-ray diffraction (PXRD), scanning electron microscopy (SEM) coupled with energy dispersive X-ray spectroscopy (EDX), and inductively coupled plasma optical emission spectroscopy (ICP-OES) methods and discussed. In vitro tests using human osteosarcoma MG-63 cell lines were performed accessing cell adhesion to the surface of the prepared Sr^2+^-substituted β-TCP ceramics and the cytocompatibility of the prepared materials.

## 2. Results and Discussion

### 2.1. Analysis of TCP and SrTCP Powders

#### 2.1.1. PXRD Study

PXRD patterns of the prepared TCP, 0.1SrTCP, and 0.5SrTCP powders are shown in Figure 1. The phase analysis revealed the presence of two types of phases in the samples: the β-TCP-type structure (β-Ca_3_(PO_4_)_2_) and the apatite-type structure (Ap, Ca_10_(PO_4_)_6_(OH)_2_) [52,54]. The Ap-type phase can be formed according to the reaction (1):(1)$$\begin{array}{c} \left(9-x\right)CaO+xSrO+6{\left({NH}_{4}\right)}_{2}{HPO}_{4}\to \\ {Ca}_{9-x}{Sr}_{x}{\left({PO}_{4}\right)}_{6}{\left(OH\right)}_{2}+12{NH}_{3}+9H_{2}O \end{array}$$
where *x* is 0, 0.1, and 0.5.

Quantitative phase analysis results obtained by the Rietveld method using the JANA2006 software (JANA, Inc., Universal City, TX, USA) and the data from the literature are given in Table 1. Based on Table 1 and Figure 1, it can be concluded that the quantities of the β-TCP and the Ap-type phases are different. 0.5SrTCP powder sample is characterized by the highest β-TCP phase amount (83 wt.%), which is consistent with the data in the literature [50,54]. The lowest content of the Ap-phase can be explained by the inhibition of its formation by Sr^2+^ ions, in accordance with [56]. The inhibition rate is rising with the increase of the Sr^2+^ concentration in the range of 0.3–10 mol.% [56]. Since 0.1SrTCP sample contains 3.33 mol.% of Sr^2+^, this value is insufficient for suppressing the growth of the Ap-type phase. The presence of the pyrophosphate phase in the β-TCP powders obtained by the mechano-chemical activation method has been reported in the literature [5,57] due to the condensation of HPO_4_^2−^ according to reaction (2):(2)$$2{Ca}^{2+}+2{HPO}_{4}^{2-}\to {Ca}_{2}P_{2}O_{7}+H_{2}O$$

Furthermore, Ca_2_P_2_O_7_ (Figure 1a) forms the β-TCP structure as it follows from the reaction (3):(3)$${Ca}_{2}P_{2}O_{7}+Ca{\left(OH\right)}_{2}\to {Ca}_{3}{\left({PO}_{4}\right)}_{2}+H_{2}O$$
where Ca(OH)_2_ is formed upon the addition of the distilled water into the initial reaction mixture (4):(4)$$CaO+H_{2}O\to Ca{\left(OH\right)}_{2}$$

Additionally, no impurities of amorphous CP or α-TCP phases were found in all the studied powder samples. Previously, it was shown that Sr^2+^ substitution in the β-TCP structures did not affect the thermal stability of the α-TCP phase and could suppress its formation for up to 10 mol.% of Sr^2+^ substitution [58].

Only a slight shift of the diffraction peaks versus lower angles with an increase in the Sr^2+^ concentration was observed (Figure 1) due to a small mol.% substitution. Usually, this shift is caused by a mismatch in the size between Ca^2+^ (r_VIII_ = 1.12 Å) and Sr^2+^ (r_VIII_ = 1.26 Å) ions [48]. Crystallographic data for 0.5SrTCP (16.67 mol.% Sr^2+^) calculated by the Le Bail decomposition are in accordance with [48] (Table 1).

PXRD pattern of 0.5SrTCP at the intermediate stage, after 400 °C heating, is shown in Figure 1b. The broad peaks are attributed to a poor crystalline Ap-phase. Further calcination leads to β-TCP stabilization (Figure 1a).

#### 2.1.2. SEM Investigation

SEM images of TCP and SrTCP powders are presented in Figure 2. The particle size of Sr^2+^-containing powders is smaller compared to the pure TCP sample and can be observed at 10 μm resolution (Figure 2a,c,e). Previously, it was shown that the presence of Sr^2+^ ions in CPs retard their crystallization process [58], so the sample with the highest Sr^2+^ concentration shows the smallest particles. The smaller particle size can be explained by a greater tendency of SrTCP powders to agglomerate [52], so the microstructure of the prepared powders is represented by agglomerates with sizes up to 40 μm for TCP powder and up to 85 μm for 0.1SrTCP and 0.5SrTCP (Figure 2b,d,f).

EDX analysis was used to confirm the chemical composition of the prepared powders (Table 2). The measurements were performed getting seven points for each sample. The Ca:Sr:P ratios were obtained, and only a slight deviation from the expected bulk composition was detected.

### 2.2. Analysis of TCP and SrTCP Ceramics

#### 2.2.1. PXRD Study

PXRD patterns of TCP, 0.1SrTCP, and 0.5SrTCP ceramics sintered at 1100 °C are shown in Figure 3. The phase compositions of the samples are close to those of the original powders calcinated at 900 °C. TCP and 0.5SrTCP ceramics contain the β-TCP as the main phase, while 0.1SrTCP is represented mainly by the Ap-phase (Figure 3). The heat treatment at 1100 °C leads to the decomposition of the Ap-phase with the formation of the β-TCP-type structure. However, no impurity of the Ca_2_P_2_O_7_ phase was detected in the samples.

#### 2.2.2. SEM Investigation

After the sintering at 1100 °C of TCP and SrTCP powders that were preheated at 400 °C, the samples of ceramics were obtained (Figure 4). With the increase in the Sr^2+^ concentration in TCP from 3.33 mol.% to 16.67 mol.%, the formation of fused particles connected to each other was observed (Figure 4a–f), which indicates the sintering process. The grain size increased with the rise in the Sr^2+^ concentration (Figure 4e,f), as was evidenced by the results obtained in [55]. Apparently, for 0.5SrTCP, a liquid phase is formed earlier and the sintering proceeds according to the liquid-phase mechanism. Previously, in [54] it was shown that TCP ceramics with 4 and 8 mol.% of Sr sintered at 1250 °C were also characterized by larger grains in comparison with pure TCP ceramics. The chemical composition of the ceramic samples is given in Table 2.

The small deviations of the Ca^2+^ and Sr^2+^ ion concentration on the surface and in the bulk composition were detected by the EDX and ICP-OES methods, respectively. Nevertheless, these results are in accordance with the expected chemical composition (Table 2).

#### 2.2.3. Mechanical Strength Measurements

The bending strength of the ceramics measured by the three-point bending method was 35 ± 3 MPa for β-TCP ceramics and 27 ± 3 MPa for 0.5SrTCP ceramics (Figure 5). The decrease in the strength of 0.5SrTCP ceramics was likely influenced by the increase in grain size when Ca^2+^ is replaced by Sr^2+^ (Figure 4a,e) according to [55]. However, the bending strength depends on a number of factors. The change in the chemical composition and ratio of TCP-type/Ap-type phases likely also contributed to the bending strength, leading to a lower value for the Sr-substituted ceramic sample compared to the pure TCP ceramic.

### 2.3. Behavior of TCP and 0.5SrTCP Ceramics in a Model Liquid

#### 2.3.1. PXRD and ICP-OES Study

The behavior of TCP and 0.5SrTCP ceramics in the 0.9% NaCl in a TRIS buffer with a pH of 7.4 was studied. The ceramic samples were soaked in a saline solution for 21 days. Figure 6 shows PXRD patterns of TCP and 0.5SrTCP ceramic samples after soaking. The TCP ceramic is characterized by an increase in the apatite-type phase quantity (Figure 6). Compared to the as-prepared TCP ceramic, in the soaked sample, the reflexes corresponding to the Ap-phase appear to be more intense. The formation of the Ap-phase is due to the dissolution of the β-TCP phase and the release of the Ca^2+^ and PO_4_^3−^ ions into the solution, which then precipitated as the Ap-phase on the surface of β-TCP [64]. Moreover, the split of the main (0 2 10) reflection (Figure 6) of the TCP sample corresponds to the formation of a metastable α-TCP phase with the main (0 3 4) reflection close to the β-TCP phase [65]. The appearance of the α-TCP phase was previously observed in [55,64,66]. Another mechanism influencing the increase in the Ap-type phase content can be explained by the hydrolysis reaction of the α-TCP phase with the formation of calcium-deficient apatite [61] (5):(5)$$3{Ca}_{3}{\left({PO}_{4}\right)}_{2}+H_{2}O\to {Ca}_{9}({HPO}_{4}){\left({PO}_{4}\right)}_{5}(OH)$$

Additionally, a smooth peak in PXRD pattern of TCP ceramic after soaking can be observed, which is attributable to the OCP phase (Figure 6). The formation of the OCP arises from the local increase in the Ca^2+^ and PO_4_^3−^ ion concentration around TCP ceramics due to the dissolution process. So, OCP can cause overgrowth in the initial β-TCP phase [67]. Additionally, it was reported that OCP is converted to the Ca-deficient Ap-phase in the supersaturated solutions (Equation (6)) [68,69]:(6)$${Ca}_{8}{\left({HPO}_{4}\right)}_{2}{\left({PO}_{4}\right)}_{4}\cdot 5H_{2}O+(2-x)Ca{\left(OH\right)}_{2}\to {Ca}_{10-x}H_{x}{\left({PO}_{4}\right)}_{6}{\left(OH\right)}_{2-x}+7H_{2}O$$
where *x* can be ranged from 0 to 2. Therefore, the transformation of the TCP in the saline solution can be described as follows: β-TCP → α-TCP, OCP → Ap.

The accumulative release amount of Ca^2+^ ions from TCP ceramic after soaking in the saline solution for 21 days was measured by the ICP-OES. The average accumulative release amount was 0.025 ± 1 g/L. This value is below the values of ionized calcium in the blood serum, which is 0.088–0.104 g/L [70]. This result is comparable to previous data on the β-TCP ceramics [22] soaked in a TRIS buffer solution, which means that the solubility of ceramics does not significantly depend on the type of solution.

The phase composition of the 0.5SrTCP sample did not change after 21 days in the saline solution, and no new phases were detected in the sample. However, in PXRD pattern of 0.5SrTCP (with respect to TCP), the split of the main diffraction reflexes is absent (Figure 6) due to the inhibition of the α-TCP phase formation by the Sr^2+^ substitution, as was shown earlier in [55,58]. Previously, it was also shown that the Ca_2_P_2_O_7_ phase retired the phase transformation: β-TCP → α-TCP [71]. Since the Ca_2_P_2_O_7_ phase is absent in all our ceramic samples, the inhibition effect is related to the Sr^2+^ ions. Additionally, 0.5SrTCP sample is characterized by a slower dissolution rate in comparison with TCP sample, since the Ap-phase was not observed in the sample after 21 days of soaking (Figure 6).

The results of the PXRD study are in agreement with the ICP-OES data. The slower dissolution rate of 0.5SrTCP sample in comparison with pure TCP ceramic was confirmed by measurements of Ca^2+^ concentration in solutions after 1, 3, and 21 days of soaking. The average accumulative release amount of Ca^2+^ from 0.5SrTCP ceramics was 0.017 ± 1 g/L in the solution soaked for 21 days. Therefore, 0.5SrTCP sample is more resistant to dissolution compared to TCP. The Sr^2+^ ions released from 0.5SrTCP after soaking for 1, 3, and 21 days are shown in Figure 7.

#### 2.3.2. SEM Investigation

According to the obtained SEM results (Figure 8), after 21 days of soaking the ceramics in the saline solution, their microstructure changed. In the case of TCP, the angular-shape particle size of the TCP increased, and the new Ap-phase formed (Figure 6 and Figure 8a,b), whereas, in the case of 0.5SrTCP, the spherical-shape particle size decreased without any phase transformation. This experimental evidence may likely be explained by the presence of Sr. The small particles observed on the surface of the samples were likely formed due to the recrystallization of the OCP phase. Additionally, some changes in the chemical composition of the TCP ceramics after soaking can be observed in Table 2. Indeed, the concentration of Ca^2+^ ions on the ceramics’ surfaces decreased, confirming that the Ca^2+^ ions released from the bulk of the ceramic samples were in accordance with the obtained ion release results.

### 2.4. Results of In Vitro Experiments

It was revealed that the rate of expansion of the human osteosarcoma MG-63 cells on the surface of ceramics after 8 days of cells’ seeding either did not differ from the control or slightly exceeded the control sample values. Thus, for 0.5SrTCP ceramic on the sixth and eighth day of the experiment, the optical density values of the formazan solution statistically significantly exceeded the ones of the control (see Table 3 and Figure 9).

Generally, a material can be considered cytocompatible if the viability of cells percentage exceeds 70%. In our case, based on the results shown in Table 3, both the ceramics—TCP and 0.5SrTCP—are cytocompatible, with the difference being that only at the first experimental time point (1 day) TCP is slightly better, while for the rest of the experiment (4, 6, and 8 days), the cytocompatibility of 0.5SrTCP ceramics is higher than that of TCP ceramic and of the control sample.

The high viability of MG-63 cells on TCP and 0.5SrTCP ceramics was also confirmed by the live/dead-cell study after 24 h of cell seeding on the prepared materials (Figure 10). From the presented images it can be observed that almost the entire population of cells, both on the control and on TCP and 0.5SrTCP ceramics, is alive, and only the presence of some single dead cells is detected (Figure 10).

Spreading and adhesion of the osteoblast-like MG-63 cells on TCP and 0.5SrTCP ceramic surfaces were evaluated after 24 h of cell seeding (Figure 11 and Figure 12). The shape and area of spreading of cells were studied by means of fluorescence microscopy. As one can observe from Figure 11 and Figure 12, the MG-63 cells vary in their shape; the cells are larger in size and of a polygonal osteoblast-like shape in the control group and smaller in size and mainly of a polygonal shape on the surfaces of the TCP and 0.5SrTCP ceramics. The area of spreading is almost twice as much as for the control sample (Figure 11). A comparison of the results obtained for TCP and 0.5SrTCP ceramics (Figure 11) shows that the cell spreading area was the same for both the ceramic materials, but the cell osteoblast-like shape (Figure 12) was more characteristic for the 0.5SrTCP ceramic.

## 3. Materials and Methods

### 3.1. Synthesis of Strontium Substituted Tricalcium Phosphate Powders and Ceramics Preparation

Sr^2+^-substituted tricalcium phosphates Ca_3−*x*_Sr*_x_*(PO_4_)_2_ (SrTCP) with *x* = 0.1 and 0.5 (3.33 mol.% and 16.67 mol.%, respectively) were synthesized using the mechano-chemical activation method, as described earlier in [5]. The raw materials of «chemical grade» purity were used: CaO (Sigma-Aldrich, St. Louis, MO, USA) and SrO (Sigma-Aldrich, St. Louis, MO, USA) calcined at 950 °C and (NH_4_)_2_HPO_4_ (Khimmed, Moscow, Russia). The chemical reaction can be described by the following Equation (7):(7)$$3-xCaO+xSrO+2{\left({NH}_{4}\right)}_{2}{HPO}_{4}\to {Ca}_{3-x}{Sr}_{x}{\left({PO}_{4}\right)}_{2}+3H_{2}O+4{NH}_{3}$$
where *x* is 0 (0 mol.%), 0.1 (3.33 mol.%), and 0.5 (16.67 mol.%). The sample with *x* = 0 (pure TCP) was used as a reference.

The stoichiometric quantities of the initial reagents were ground in a planetary mill in a Teflon drum with 200 g of zirconium oxide grinding bodies at a rotation speed of 1500 min^−1^ for 30 min. After that, 200 mL of distilled water was added to the drum, and the grinding was continued for another 30 min. Then, the resulting suspension was filtered in a Buchner funnel, washed with distilled water, and dried at 110 °C for 12 h. The resulting products were preheated at 400 °C to remove water and ammonia residues and then annealed at 900 °C for 2 h to obtain TCP and SrTCP powders.

For ceramic samples preparation, the preheated at 400 °C powders were first pressed in a steel mold under 100 kg/cm^2^ and then sintered in a furnace at 1100 °C for 2 h.

### 3.2. PXRD Analysis

PXRD patterns were obtained on Thermo ARL X’TRA powder diffractometer (Bragg–Brentano geometry, Scintillator detector, CuKα radiation, λ = 1.5418 Å, Thermo Fisher Scientific, Waltham, MA, USA). The PXRD data were collected at the 5°–65° 2theta range, with a 0.02° step. The PXRD experiments were performed at room temperature. The Le Bail decomposition [72] was applied in the JANA2006 software (JANA, Inc., Universal City, TX, USA) [73] for the refinement of the unit cell parameters and the volume of the synthesized powders. The phase analysis was carried out by means of the Crystallographica Search-March program (version 2.0.3.1) and the JCPDS PDF#2 and PDF#4 databases. The Rietveld method using the JANA2006 software (JANA, Inc., Universal City, TX, USA) was applied for the determination of the β-TCP, Ap- and Ca_2_P_2_O_7_ phase quantities in the investigated powder samples. Crystallographic data of space group (SG), unit cell, and atomic coordinates of β-Ca_3_(PO_4_)_2_ (PDF#2 No 70-2065) [59], Ca_10_(PO_4_)6(OH)_2_ (ICDD 183744, PDF#2 73-1731) [74], and Ca_2_P_2_O_7_ (PDF#4 No 04-009-3876) [75] were used as the initial parameters. The fifteenth-order polynomial fit was applied to refine the background and the modified Pseudo-Voigt function (the peak profiles). The atomic coordinates were fixed as in the card (PDF#2 or PDF#4), whereas the unit cell parameters were refined.

### 3.3. SEM Investigation

SEM observations of the powder samples as well as TCP, 0.1SrTCP, and 0.5SrTCP ceramics were performed using a Tescan VEGA3 (Tescan, Czech) scanning electron microscope equipped with an Oxford Instruments X-Max 50 silicon drift energy-dispersive X-ray spectrometry (EDXs) system with AZtec (Oxford Instruments NanoAnalysis, France) and INCA software (JANA, Inc., Universal City, TX, USA) (Base Product package). Samples were coated with a thin layer of carbon for the SEM examinations. SEM images were acquired using secondary electron and backscattered electron imaging techniques. The EDX analysis results were based on the Ca_K_, Sr_K_, and P_K_ edge lines. The oxygen content was not quantified by EDX.

### 3.4. Three-Point Bending Method

The strength of 5 ceramic samples with a 4 × 4 × 40 mm size of each type was measured by the three-point bending method, as described in [76]. An Instron 3382 (Instron Corp. ElectroPuls E3382, Norwood, MA, USA) mechanical testing machine with the speed of movement of the movable traverse of 5 mm/min was applied. The press head and the two reference points were rounded to avoid shear loading and cutting. The sample was positioned horizontally, centered on the supports, and the clamping force was directed vertically to the middle part of the sample. Each sample was loaded at a constant rate of 0.05 mm/sec until destruction. The strength of the samples was determined as the maximum load during destruction.

### 3.5. Solubility of SrTCP Ceramics in Model Liquid

The solubility of ceramics in saline solution was investigated by following the concentration of Ca^2+^ ions. The composition of the saline solution was 0.9% NaCl in a TRIS buffer with a pH of 7.4. The sintered ceramic (1 g) was placed in a container with 50 mL of saline solution and placed in a thermostat at a physiological temperature of 37 °C for 21 days. The accumulative release amount of Ca^2+^ ions was measured using inductively coupled plasma optical emission spectroscopy (ICP-OES, 720-ES axial spectrometer (Agilent Technologies, NY, USA)). The obtained data were reported as mean ± standard deviation.

### 3.6. In Vitro Investigation of MG-63 Cell Adhesion and Cytocompatibility of SrTCP Ceramic Surface

To evaluate the adhesion of the standard human osteosarcoma MG-63 cell line (ThermoFisher, Waltham, MA, USA) to the surface of pure TCP ceramics and samples containing Sr, the area of cell culture after 24 h of cultivation on samples and on polystyrene (control sample) was determined. For this purpose, sterilized granules of the prepared materials were placed in the wells of a 24-well plate (Costar). After that, the cell suspension with a density of 15 × 10^3^ cells/cm^2^ was added to each well containing the ceramic samples and the control.

After 24 h, the cells adhered to the granules were fixed with 4% formaldehyde solution for 10 min at room temperature and then washed three times with phosphate-buffered saline (PBS) (Sigma-Aldrich, St. Louis, MO, USA). Then, the cells were treated with 0.05% Triton X-100 (TX100) for 10 min at room temperature and washed three times in PBS.

The cytoskeleton was stained with the Falloidin-iFluor 488 (Abcam, Waltham, MA, USA) diluted in the ratio of 1:1000. The cells were incubated for 1.5 h at room temperature, according to the manufacturer’s recommendation.

After washing the cells from unbound dye, they were stained with nuclei dye DAPI (Invitrogen, Waltham, MA, USA).

Images of human MG-63 sarcoma cells, stained by phalloidin conjugated to the Alexa Fluor 488 and DAPI, were obtained by means of a confocal system AIR, installed on the microscope Nikon Ti (Nikon, Japan). A Water Immersion Nikon CFI Plan Fluor 20× MI mm DIC N2 × 0.75 NA lens (Nikon, Japan) was used. Wavelengths of excitation/emission were ex = 405 nm/em 450 (50) nm (DAPI) and ex = 488 nm/em 525 (50) nm (FITC, Alexa Fluor 488). The images were processed by means of the NIS Elements ver. 4.51 (Nikon, Japan) program.

The cytocompatibility of the obtained ceramic samples was accessed by direct contact of the test cell culture—MG-63 human osteosarcoma cells—with the surface of investigated ceramics. Sterile ceramic samples were placed in a 24-well plate (Costar, Antioch, CA, USA) (4 samples of each type of material for each experimental duration: 3 of them for cytocompatibility investigation and 1 blank) and the cell suspension was added at a concentration of 40.0 × 10^3^ cells/cm^2^ for each well (the seeding density was 20.0 × 10^3^ cells/cm^2^). Cell cultivation was carried out for 1, 4, 6, and 8 days with a regular (twice-a-week) change of culture medium. Wells with cells growing on polystyrene served as controls. Cell viability at different stages of cultivation was tested by the MTT method. The optical density of formazan solution (formazan is a product of MTT-tetrazolium reduction) was measured by means of a spectrophotometer Multiscan (Thermoscientific, Waltham, MA, USA) at a wavelength of 540 nm. For each sample of investigated ceramic material at a certain time, the percentage of cell viability (PVC) was calculated using the following equation:VC = (mean OD sample/mean OD blank) × 100%

OD—the value of optical density of the formazan solution in the experiment and in the negative control, respectively.

## 4. Conclusions

The TCP, 0.1SrTCP, and 0.5SrTCP powders were prepared by applying the mechano-chemical activation method. The corresponding ceramic samples were obtained by pressing and sintering powders at 1100 °C. The β-TCP was the main phase in TCP and 0.5SrTCP powders, while the apatite phase was predominant in the 0.1SrTCP powder. The microstructure of the prepared powders was represented by agglomerates with sizes up to 40 μm for TCP and up to 85 μm for 0.1SrTCP and 0.5SrTCP.

The phase composition of the prepared ceramics was close to that of the original powders calcinated at 900 °C. TCP and 0.5SrTCP ceramics contained β-TCP as the main phase, while 0.1SrTCP was represented mainly by the Ap-phase. The heat treatment at 1100 °C led to the decomposition of the Ap-phase with the formation of β-TCP.

The bending strength was 35 ± 3 MPa for TCP ceramic and 27 ± 3 MPa for 0.5SrTCP ceramic.

After soaking in 0.9% NaCl in a TRIS buffer for 21 days, TCP ceramic was characterized by an increase in the Ap-type phase quantity, while the composition of 0.5SrTCP ceramic did not change.

The dissolution behavior was different for the prepared ceramics. TCP ceramic showed higher solubility, which was confirmed by the higher Ap-phase content after soaking, compared to the 0.5SrTCP ceramic. These results were confirmed by the ICP-OES measurements. The average accumulative release amount of Ca^2+^ from 0.5SrTCP ceramics was 0.017 ± 1 g/L, while for TCP, the measured value was 0.025 ± 1 g/L, showing higher resistance to dissolution for 0.5SrTCP. The release of Sr^2+^ ions in the solution increased with time and reached 0.84 mg/L after 21 days of soaking. The impurities of OCP and α-TCP phases were found in TCP ceramic accounting for the transformation according to the scheme: β-TCP → α-TCP, OCP → Ap. The 0.5SrTCP ceramic showed higher resistance to dissolution, and the incorporation of Sr^2+^ into the β-TCP host structure suppressed the formation of the metastable α-TCP phase. According to SEM results, the particle size in 0.5SrTCP ceramic decreased after soaking, compared to TCP, which was also caused by a higher solubility of TCP resulting in the formation of large particles of the Ap-phase.

The cytocompatibility tests using the MG-63 cell line on the prepared TCP and 0.5SrTCP ceramics demonstrated that both ceramics are cytocompatible, but the cytocompatibility of 0.5SrTCP ceramic is higher than that of TCP and of the control. High viability of cells on both ceramics was also detected. Partial release of Sr^2+^ ions into the solution led to the retention of an osteoblast-like cell shape and improved the cytocompatibility. Moreover, the results obtained on cell adhesion and spreading evaluated after 24 h of seeding allowed us to conclude that the prepared 0.5SrTCP ceramic material is promising for further study in model experiments in vivo.

## Figures and Tables

**Figure 1 molecules-27-06085-f001:**
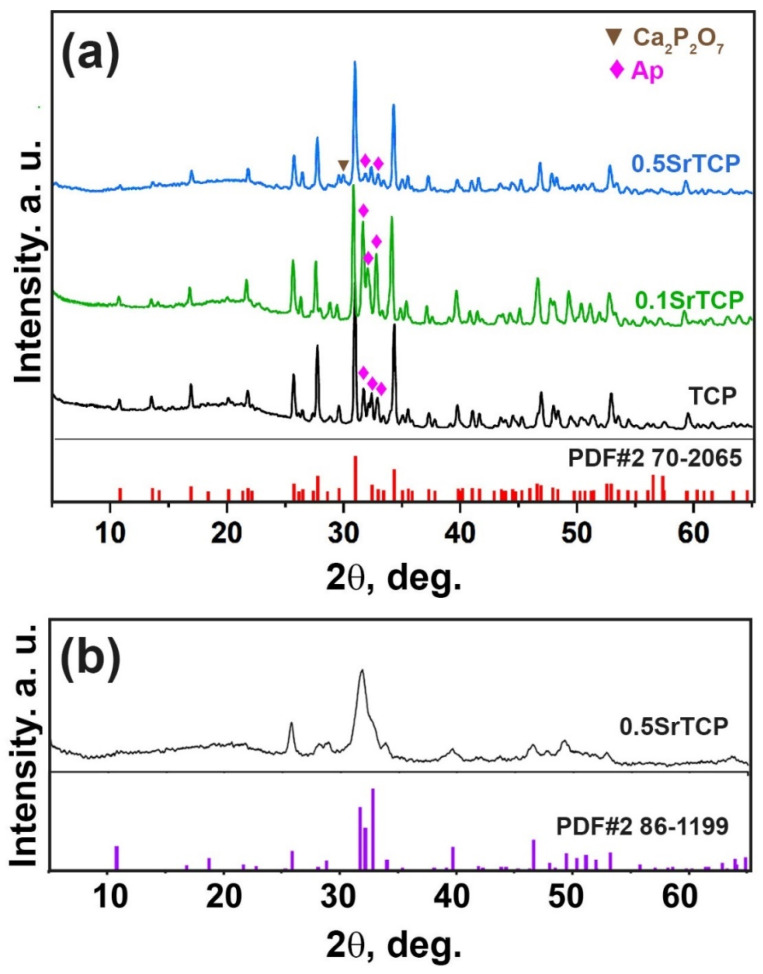
(**a**) PXRD patterns of TCP, 0.1SrTCP, and 0.5SrTCP powders along with PDF#2 card 70-2065 (β-Ca_3_(PO_4_)_2_) as a reference. The impurities of the Ap-type phase and calcium pyrophosphate (Ca_2_P_2_O_7_) are labeled. (**b**) PXRD pattern of 0.5SrTCP powder preheated at 400 °C.

**Figure 2 molecules-27-06085-f002:**
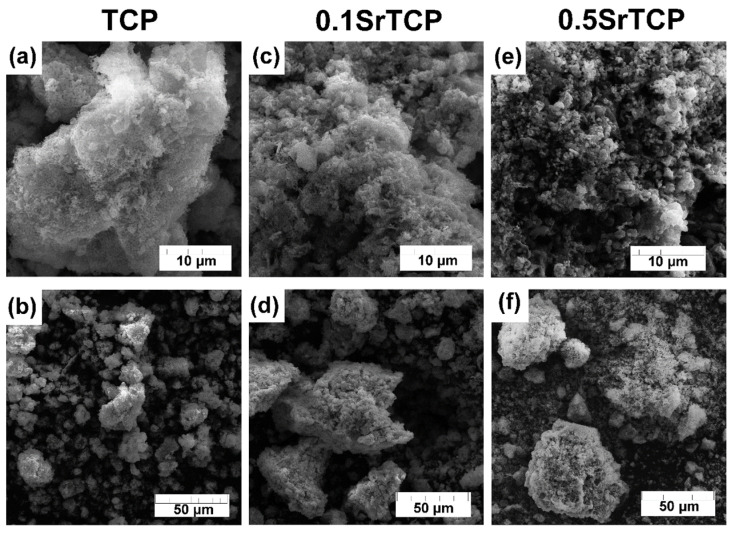
SEM images of TCP (**a**,**b**), 0.1SrTCP (**c**,**d**), and 0.5SrTCP (**e**,**f**) powders calcinated at 900 °C at 10 and 50 μm resolutions.

**Figure 3 molecules-27-06085-f003:**
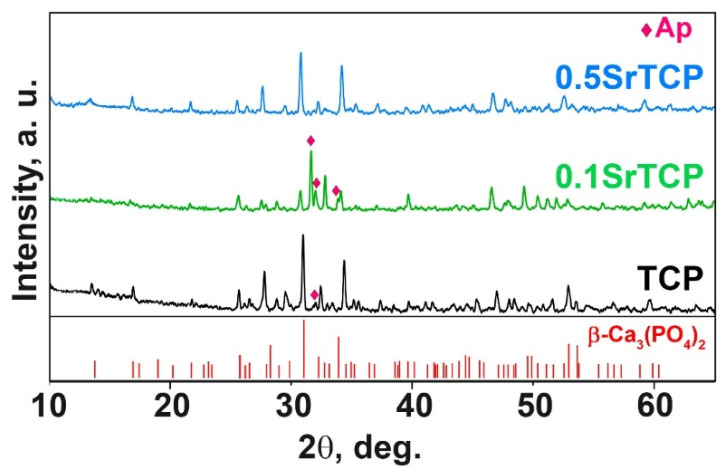
XRD patterns of TCP, 0.1SrTCP, and 0.5SrTCP samples calcined at 1100 °C along with PDF#2 card No 70-2065 (β-Ca_3_(PO_4_)_2_).

**Figure 4 molecules-27-06085-f004:**
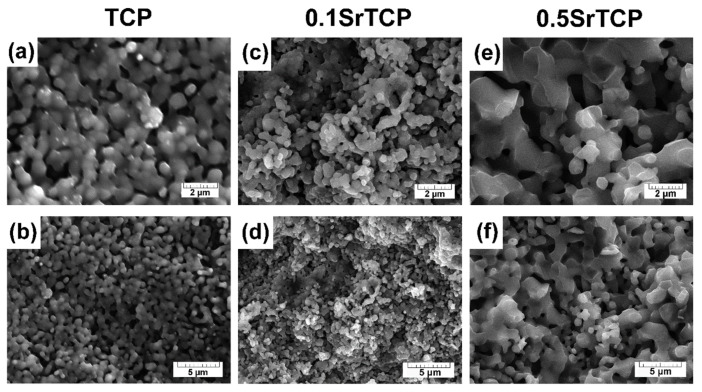
SEM images of TCP (**a**,**b**), 0.1SrTCP (**c**,**d**), and 0.5SrTCP (**e**,**f**) ceramics sintered at 1100 °C at 2 and 5 μm resolution.

**Figure 5 molecules-27-06085-f005:**
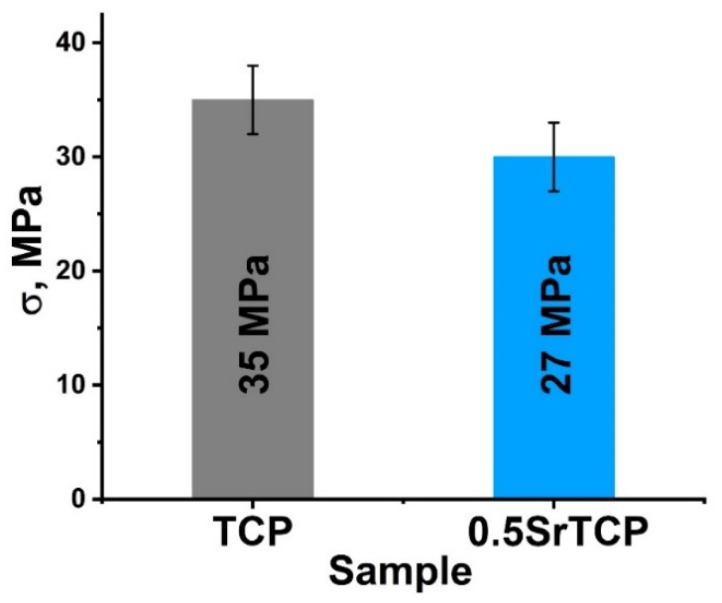
Mechanical strength of TCP and 0.5SrTCP ceramics sintered at 1100 °C.

**Figure 6 molecules-27-06085-f006:**
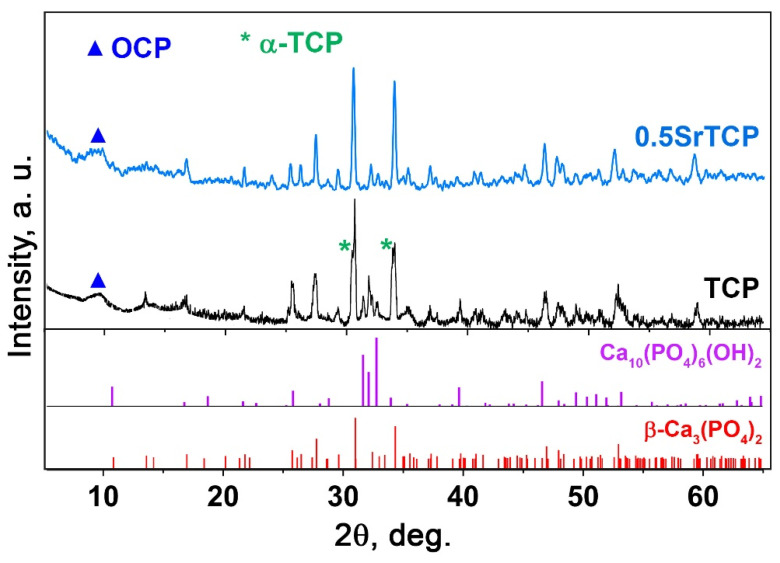
PXRD patterns of TCP and 0.5SrTCP ceramics calcined at 1100 °C and kept in the saline solution for 21 days.

**Figure 7 molecules-27-06085-f007:**
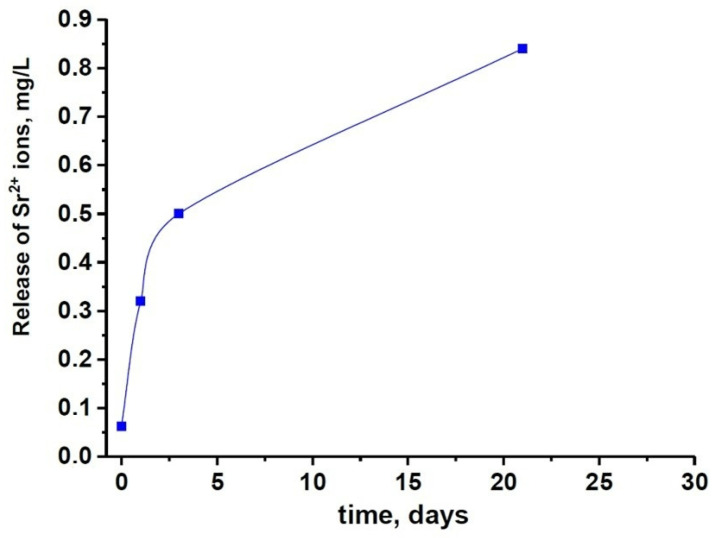
The solubility of 0.5SrTCP ceramic in the saline solution.

**Figure 8 molecules-27-06085-f008:**
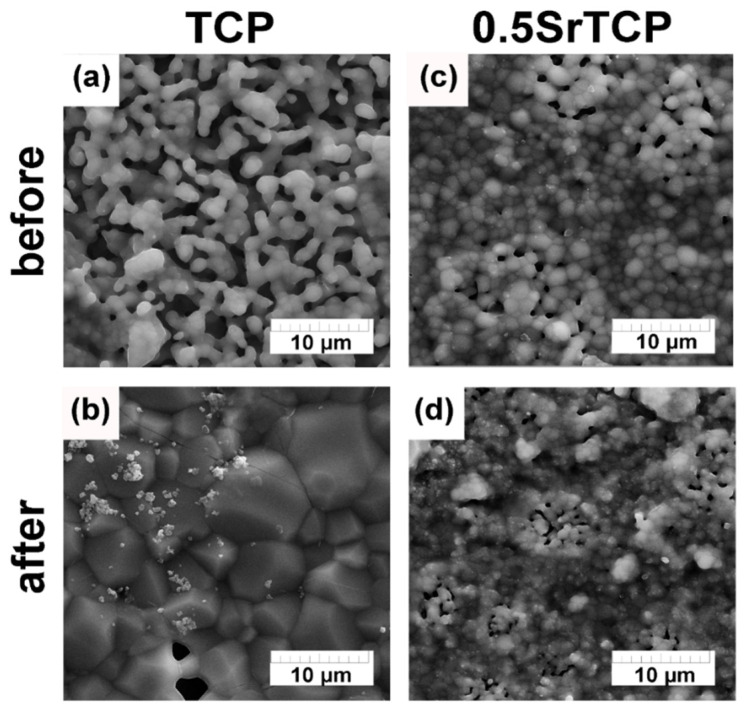
SEM images of TCP and 0.5SrTCP ceramic microstructure before (**a**,**b**) and after soaking in saline solution for 21 days (**c**,**d**).

**Figure 9 molecules-27-06085-f009:**
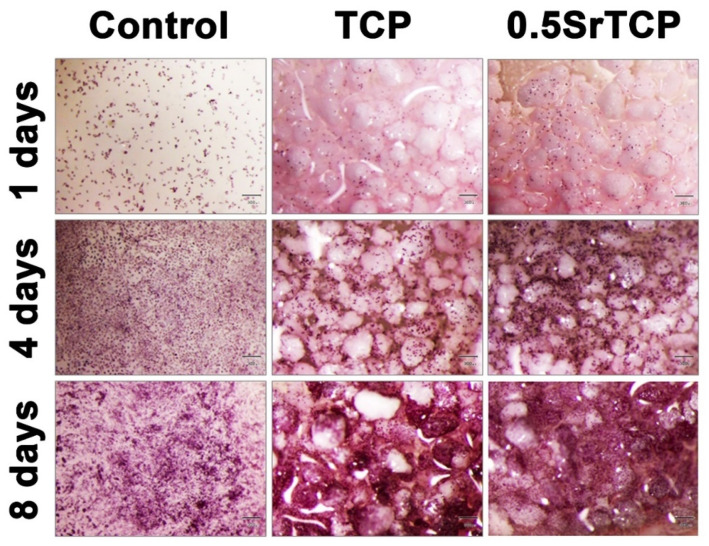
Population of MG-63 cells grown on control and on TCP and 0.5SrTCP ceramics. The scale bars correspond to 300 μm.

**Figure 10 molecules-27-06085-f010:**
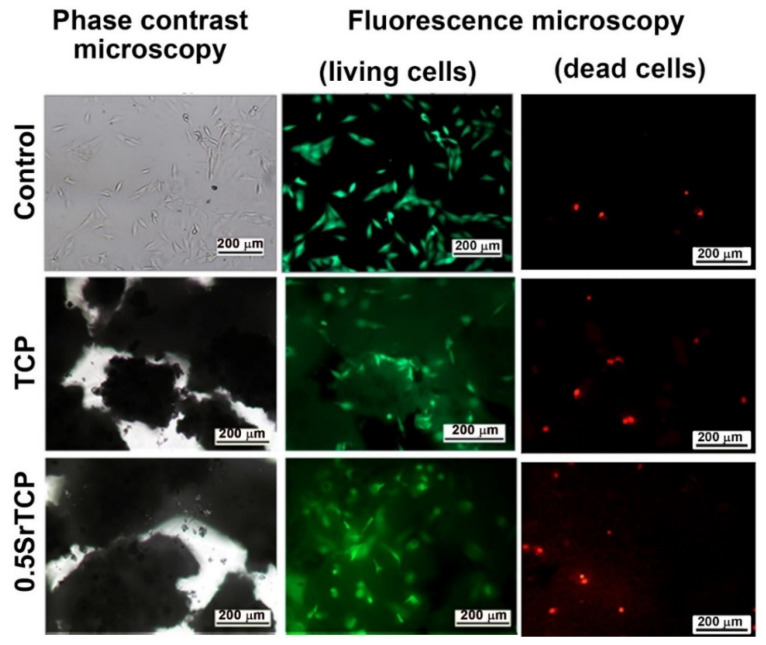
Fluorescence microscopy of living (**green**) and dead (**red**) MG-63 cells in direct contact with TCP and 0.5SrTCP ceramics (live/dead assay, 24 h of cell growth).

**Figure 11 molecules-27-06085-f011:**
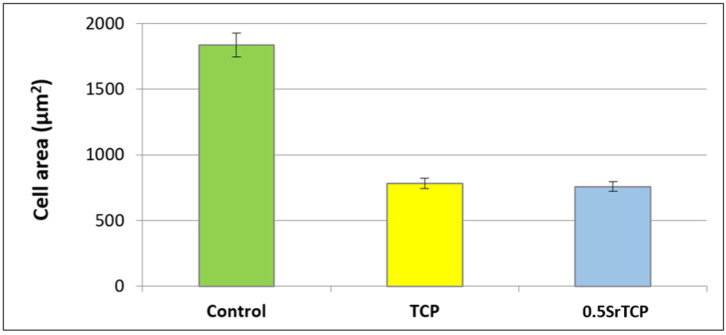
Spreading of MG-63 cells on control, TCP, and 0.5SrTCP ceramics (24 h of cell growth).

**Figure 12 molecules-27-06085-f012:**
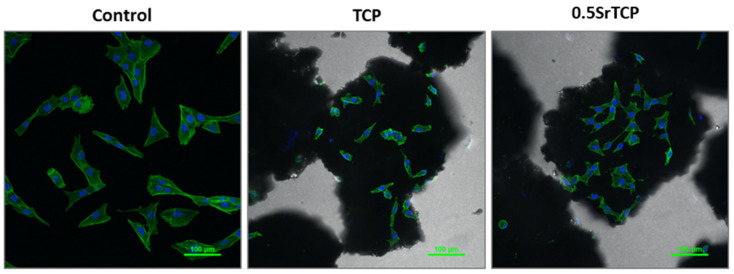
Confocal fluorescence microscopy of MG-63 cells in direct contact with TCP and 0.5SrTCP ceramics (24 h of cell growth, monochrome images of FITC conjugated Phalloidin (**green**) and DAPI (nuclei, **blue**)).

**Table 1 molecules-27-06085-t001:** Quantitative phase analysis of TCP and SrTCP powders and the corresponding reference data.

Sample	Phase	wt%, Jana 2006	Space Group	Unit Cell Parameters		Reference
				*a*, Å	*c*, Å	
TCP	β-TCP Apatite	79 21	*R*3*c* *P*6_3_/*m*	10.416(6) 9.416 (9)	37.385(2) 6.878(9)	This work
0.1SrTCP	β-TCP Apatite	52.5 47.5	*R*3*c* *P*6_3_/*m*	10.478(3) 9.434(4)	37.483(1) 6.902(5)	This work
0.5SrTCP	β-TCP Apatite Ca_2_P_2_O_7_	83 11 6	*R*3*c* *P*6_3_/*m* *P*4_1_	10.482(3) 9.441(4) 6.687(9)	37.491(2) 6.903(4) 24.146(8)	This work
β-Ca_3_(PO_4_)_2_	β-TCP	100	*R*3*c*	10.435(3)	37.403(7)	[59]
Ca_2.9_Sr_0.1_(PO_4_)_2_	β-TCP	100	*R*3*c*	10.448(4)	37.409(7)	[60]
Ca_10_(PO_4_)_6_(OH)	Apatite	100	*P*6_3_/*m*	9.423(2)	6.883(8)	[61]
Ca_9.92_Sr_0_._08_(PO_4_)_6_(OH)_2_	Apatite	100	*P*6_3_/*m*	9.435(3)	6.889(4)	[62]
Sr_10_(PO_4_)_6_(OH)_2_	Apatite	100	*P*6_3_/*m*	9.745(1)	7.265(3)	[63]

**Table 2 molecules-27-06085-t002:** Chemical composition of TCP and SrTCP samples obtained by EDX and ICP-OES data. The values are given with a relative error of 0.05%.

	Expected Formula	Nominal $\frac{n\left(Sr^{2+}\right)\times 100\%}{n\left(Ca^{2+}+Sr^{2+}\right)}$	Calculated Formula from EDX Data	EDX Data $\frac{n\left(Sr^{2+}\right)\times 100\%}{n\left(Ca^{2+}+Sr^{2+}\right)}$	Calculated Formula from ICP-OES Data	ICP-OES Data $\frac{n\left(Sr^{2+}\right)\times 100\%}{n({Ca}^{2+}+{Sr}^{2+})}$	Molar Ratio from EDX Data $\frac{Ca^{2+}+Sr^{2+}}{P}$
Powder samples							
TCP	Ca_3_(PO_4_)_2_	0 mol.% Sr^2+^	Ca_2.96_(PO_4_)_2_	0 mol.% Sr^2+^	n/a	n/a	1.48
0.1SrTCP	Ca_2.9_Sr_0.1_(PO_4_)_2_	3.33 mol.% Sr^2+^	Ca_2.68_Sr_0.21_(PO_4_)_2_	7.27 mol.% Sr^2+^	Ca_2.89_Sr_0.11_(PO_4_)_2_	3.66 mol.% Sr^2+^	1.45
0.5SrTCP	Ca_2.5_Sr_0.5_(PO_4_)_2_	16.67 mol.% Sr^2+^	Ca_2.44_Sr_0.42_(PO_4_)_2_	17.13 mol.% Sr^2+^	Ca_2.48_Sr_0.52_(PO_4_)_2_	17.33 mol.% Sr^2+^	1.43
Ceramic samples							
TCP	Ca_3_(PO_4_)_2_	0 mol.% Sr^2+^	Ca_2.83_(PO_4_)_2_	0 mol.% Sr^2+^	n/a	n/a	1.415
0.1SrTCP	Ca_2.9_Sr_0.1_(PO_4_)_2_	3.33 mol.% Sr^2+^	Ca_2.59_Sr_0.26_(PO_4_)_2_	9.12 mol.% Sr^2+^	Ca_2.89_Sr_0.11_(PO_4_)_2_	3.66 mol.% Sr^2+^	1.425
0.5SrTCP	Ca_2.5_Sr_0.5_(PO_4_)_2_	16.67 mol.% Sr^2+^	Ca_2.55_Sr_0.39_(PO_4_)_2_	13.26 mol.% Sr^2+^	Ca_2.48_Sr_0.52_(PO_4_)_2_	17.33 mol.% Sr^2+^	1.47
Ceramic samples after soaking							
TCP	Ca_3_(PO_4_)_2_	0 mol.% Sr^2^	Ca_2.62_(PO_4_)_2_	0 mol.% Sr^2+^	n/a	n/a	1.31
0.5SrTCP	Ca_2.5_Sr_0.5_(PO_4_)_2_	16.67 mol.% Sr^2+^	Ca_2.11_Sr_0.39_(PO_4_)_2_	15.60 mol.% Sr^2+^	n/a	n/a	1.25

**Table 3 molecules-27-06085-t003:** Optical density values of the formazan solution (MTT test) and the percentage of viable cells (PVC) during the growth of MG-63 cells on TCP and 0.5SrTCP ceramic samples.

Samples	OD Value (a.u.) and PVC (%) Days			
	1	4	6	8
control	0.253 ± 0.014 100.0	0.973 ± 0.022 100.0	1.387 ± 0.036 100.0	1.991 ± 0.021 100.0
TCP	0.281 ± 0.007 111.0	0.906 ± 0.019 93.1	1.285 ± 0.038 92.6	1.881 ± 0.065 94.5
0.5SrTCP	0.252 ± 0.015 99.1	1.013 ± 0.052 104.6	1.626 ± 0.029 * 117.2	2.218 ± 0.036 * 111.4

* statistically significant difference with control (*p* < 0.05).

## Data Availability

The data are available upon an official and reasonable request.

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
