# Peer review of "Strontium Substituted β-Tricalcium Phosphate Ceramics: Physiochemical Properties and Cytocompatibility"

_molecules, 2022, doi:10.3390/molecules27186085_

Round 1

Reviewer 1 Report

On page 6, lines 189-191: "With the increase in the Sr2+ concentration in
TCP from 3.33 mol.% up to 16.67 mol.%, the formation fused particles connected to each other was observed (Figure 4, a, b, e, f), which indicates for the sintering process." If Sr effect is being mentioned, why not mentioning Figs. 4 c, d?

At 2.2.3, is the different on the values of bending strength statistically significant?

For porous ceramic samples, Weibull statistics would be more suitable.

The auhors state that the difference in bending strength is due to the difference in grain size.  This is not possible for porous samples with different pore size and pore size distribution.

On page 10, 2.3.2. SEM investigation, what is shown in Figs. 7a and 7b seems to be a phase transformation, and not a simple particle size increase.

Reviewer 2 Report

The manuscript entitled "Strontium substituted β-tricalcium phosphate ceramics: physicochemical properties and cytocompatibility" is well prepared and ready for publication.

If it is not required by the Molecules journal I would replace the Materials and Methods section after Introduction. Furthermore, I could not find information about the three-point bending method in this section. It should be added.

Authors should also add information about the size of the sinters used for mechanical tests and personally, I would recommend smoothing XRD patterns using software like Match! Crystal Impact or other software available commercially. 

Reviewer 3 Report

1. Sr2+ ions into the β-TCP improved cell adhesion, proliferation and cytocompatibility is very important in research point of view. Kindly also provide the ion release profile of the Sr2+ ions. ICP-MS should be perfomed to determine the release kinetics of Sr2+ ions and correlate the same with the cell adhesion, proliferation and cytocompatibility.

2. ROS profile of Sr2+ ions effecting the biological activity should be analyzed to solidify the impact of the Sr2+ ions.

3. It would be also interesting to mention the optical properties of the materials after inclusion of Sr2+ ions.

Round 2

Reviewer 1 Report

The paper is now ready for publication.  I agree with the provided corrections.